# The Effectiveness of Nutrition Interventions Combined with Exercise in Upper Gastrointestinal Cancers: A Systematic Review

**DOI:** 10.3390/nu13082842

**Published:** 2021-08-18

**Authors:** Fatemeh Sadeghi, David Mockler, Emer M. Guinan, Juliette Hussey, Suzanne L. Doyle

**Affiliations:** 1Discipline of Physiotherapy, Trinity College Dublin, D08 W9RT Dublin, Ireland; Sadeghif@tcd.ie (F.S.); jmhussey@tcd.ie (J.H.); 2John Stearne Medical Library, Trinity Centre for Health Sciences, D08 W9RT Dublin, Ireland; david.mockler@tcd.ie; 3Trinity College Dublin, School of Medicine, D02 R590 Dublin, Ireland; emguinan@tcd.ie; 4School of Biological and Health Sciences, Technological University Dublin, Grangegorman, D07 XT95 Dublin, Ireland

**Keywords:** cancer, gastrointestinal, nutrition, exercise, rehabilitation, multidisciplinary, body composition

## Abstract

Malnutrition and muscle wasting are associated with impaired physical functioning and quality of life in oncology patients. Patients diagnosed with upper gastrointestinal (GI) cancers are considered at high risk of malnutrition and impaired function. Due to continuous improvement in upper GI cancer survival rates, there has been an increased focus on multimodal interventions aimed at minimizing the adverse effects of cancer treatments and enhancing survivors’ quality of life. The present study aimed to evaluate the effectiveness of combined nutritional and exercise interventions in improving muscle wasting, physical functioning, and quality of life in patients with upper GI cancer. A comprehensive search was conducted in MEDLINE, EMBASE, Web of Science, Cochrane Library, and CINHAL. Of the 4780 identified articles, 148 were selected for full-text review, of which 5 studies met the inclusion criteria. Whilst reviewed studies showed promising effects of multimodal interventions on physical functioning, no significant differences in postoperative complications and hospital stay were observed. Limited available evidence showed conflicting results regarding the effectiveness of these interventions on preserving muscle mass and improving health-related quality of life. Further studies examining the impact of nutrition and exercise interventions on upper GI patient outcomes are required and would benefit from reporting a core outcome set.

## 1. Introduction

Upper GI cancers, namely esophageal cancer, gastric cancer, hepatobiliary cancers, and pancreatic cancer, were reported to affect 3.12 million individuals (17.3% of the global cancer incidence) in 2018 [1]. Adding to this, upper GI cancers were responsible for 2.6 million deaths worldwide in 2018. The treatment for upper GI cancer includes surgery, chemotherapy, and radiotherapy. Although advances in these cancer treatments, and early cancer diagnosis, has led to higher survival rates, the treatments are accompanied by adverse effects on patients’ nutritional status and physical function [2,3,4,5,6].

Cachexia and muscle loss are one of the main complications in cancer patients that hinder cancer treatment and survival [3]. Muscle wasting is accompanied by reduced physical function, fatigue, chemotherapy toxicity, lower quality of life, and a higher rate of postoperative complications [7,8,9,10]. Due to the tumor location and the unique influences of curative surgery on a patient’s dietary intake, patients with upper GI cancer are at a greater risk of malnutrition and its related complications [11,12,13]. As the survival rates for upper GI cancer continue to improve, the long-term nutritional and physical status of upper GI cancer survivors requires further attention. There is a need to offer effective care plans to prevent muscle wasting and optimize nutritional and functional status in upper GI cancer patients, which subsequently would improve treatment outcomes and enhance survivors’ quality of life.

Multimodal interventions consisting of nutrition and exercise prescription have been reported to favorably change health-related outcomes in cancer patients, such as fatigue, quality of life, and functional capacity [14,15]. These multimodal interventions may play an even more significant role in older cancer patients who are at a further increased risk of suboptimal nutritional and functional status [16].

As both nutrition and exercise have positive effects on muscle loss, combining nutrition and exercise may further improve muscle protein synthesis and increase muscle mass [17]. It should be mentioned that the impact of combined nutritional care with physical exercise specifically on body composition is not clear yet [18,19,20,21,22,23]. Thus, the present systematic review of combined nutritional and exercise interventions in upper GI cancer patients aimed to determine whether these interventions are an effective approach for preserving muscle mass. In addition, these care programs need to be evaluated to define optimal intervention design and optimal timepoint for delivery within the cancer trajectory, i.e., prehabilitation or rehabilitation. Moreover, patients’ acceptance and adherence to these programs need to be reviewed, as the effectiveness and acceptance of nutritional and exercises interventions may be different in patients with upper GI cancer compared to other cancer types, owing to the impact of surgery on food intake and postoperative complications, such as food intolerance and malabsorption.

To the authors’ knowledge, this is the first systematic review to research the effectiveness of combined nutritional and exercise interventions on muscle wasting and quality of life in upper GI cancer survivors.

The primary objective of this systematic review was to evaluate the effectiveness of nutritional interventions combined with exercise in improving outcomes in upper GI cancer patients. Specifically, it assessed whether these types of interventions, as compared to usual/standard care, can significantly prevent or reverse muscle wasting and functional decline, improve health-related quality of life, and decrease treatment complications. Moreover, this systematic review aimed to discover the acceptability of these interventions in patients diagnosed with upper GI cancer.

## 2. Materials and Methods

### 2.1. Study Registration

This study is registered with the PROSPERO database (registration number: CRD42021239675) and has been reported using the Preferred Reporting Items for Systematic Reviews and Meta-Analyses (PRISMA) checklist [24].

### 2.2. Study Eligibility Criteria

The following criteria were considered for selecting eligible studies.

### 2.3. Study Designs

All types of randomized clinical trials (RCTs), excluding feasibility and pilot studies, were included. Cross-sectional, cohort, case-control, and case report studies were excluded. Although review and systematic review articles were excluded, reference lists and citations of relevant reviews were screened for locating additional relevant studies.

### 2.4. Publication Type

Original studies published in peer-reviewed journals that were reported in English were considered eligible.

### 2.5. Participants

Studies that recruited adults (aged ≥ 18 y) diagnosed with upper GI cancer, namely esophageal cancer, gastric cancer, hepatic cancer, pancreatic cancer, and cancer of the biliary system, were included, excluding studies that enrolled patients receiving interventions as part of a palliative care plan.

### 2.6. Intervention

Studies investigating the effects of multidisciplinary care programs combining nutritional and exercise intervention were included. Studies that provided any types of nutritional and exercise interventions were included. No limitation on delivery mode, minimum or maximum intervention period, follow up, and type of setting that studies were conducted in was applied.

### 2.7. Comparators

Studies that considered standard or usual care provided to patients as the control group were included.

### 2.8. Outcomes

The following outcomes were analyzed and graded in the present systematic review if reported in the included studies.

Body composition (fat mass, fat-free mass, muscle mass) and anthropometric measurements (weight, BMI, waist/hip circumference, triceps skinfold).Health-related quality of life (HRQOL).Functional outcomes (handgrip strength, exercise capacity, and physical activity level).Dietary intake.Post-operative complications.

### 2.9. Search Strategy and Information Sources

A comprehensive literature search was conducted to identify all published and unpublished studies. Electronic databases, including MEDLINE, EMBASE, Web of Science, Cochrane Library, and CINHAL, were searched up to 26 August 2020.

The specialized search strategy was designed by the subject librarian (DM) with no limitation on type of studies, language, study design, time frame etc. The full search strategy is included as Appendix A [25]. The search result that was achieved through this search strategy was reviewed by authors (FS) to confirm that an acceptable portion of relevant studies had been retrieved by using it.

### 2.10. Data Management

All search results were imported to Covidence software (www.covidence.org) to facilitate the review process. Duplicates were excluded and multiple reports of one study were collated.

### 2.11. Selection Process

Authors SD and FS independently screened the titles and abstracts. Subsequently, SD and FS reviewed full texts independently to identify studies that met the inclusion criteria. Reasons for exclusion of the ineligible studies were recorded. Any disagreements in screening and full-text review were resolved through discussion.

### 2.12. Data Collection

FS performed data extraction independently. Extracted data included study title; publication date; correspondent author’s name and contact; study design; setting or location; sample size; participant details, including age, gender, cancer type, and treatment; and intervention design, including details of the nutrition and exercise components, mode of delivery, duration of intervention, comparators, adherence rate to intervention, withdrawal rate, adverse events, and reported outcomes (body composition and anthropometric measurements, quality of life, physical functioning, post-operation complications).

### 2.13. Risk of Bias in Individual Studies

The revised Cochrane Risk of Bias tool (RoB2) was used to determine the risk of bias [26]. SD and FS independently assessed the risk of bias for each included study. Disagreements were resolved by discussion.

### 2.14. Measuring the Intervention Effect

Extracted continuous data are presented as mean difference (MD) and standard deviation (SD) or parameter estimate (β) and confidence interval (CI). For dichotomous data (such as adverse events, complications, etc.), the treatment effect is reported as number and percent.

### 2.15. Data Synthesis

A meta-analysis was not appropriate due to the heterogeneity of studies and a high degree of variance in interventions, study period, and measured outcomes observed in the included studies. A narrative summary of the results has been provided instead.

## 3. Results

A total of 4780 articles were identified. After removing duplicates and screening titles and abstracts, 148 articles were selected for full-text review, of which 5 studies met the inclusion criteria [15,22,27,28,29]. A detailed record of the selection process and PRISMA flow diagram is presented in Figure 1. An overview of the included studies is provided in Table 1. High variation in intervention design and standard care was observed in the included studies and a detailed description of the interventions is provided in Table 2.

With regards to sample size calculation, four of the included studies conducted a power calculation (80% power) to estimate the minimum required sample size [15,22,27,29]. One of the studies failed to conduct a power calculation before starting the trial although post hoc analysis showed a power of 83–100% for the primary outcomes except lean muscle mass, for which 48% power was indicated to detect group differences [28]. Reported recruitment rates ranged from 40.3% to 96% and the dropout rates ranged from a minimum of 5% to a maximum of 25%. The adherence rate ranged from 32% to 94%. None of the studies reported major adverse events related to the intervention. A summary of the results is presented in Table 3. The result of the risk of risk bias assessment is shown in Figure 2.

### 3.1. Health-Related Quality of Life

The effects of combined nutrition interventions with physical training in improving health-related quality of life (HRQOL) were investigated in only two studies with inconsistent results [22,29]. Both studies used the European Organization for the Research and Treatment of Cancer-Core Quality of Life Questionnaire (EORTC-QLQ-C30) for measuring changes in HRQOL and the esophageal cancer-specific quality of life questionnaire (EORTC QLQ-OES18) was additionally used by Yu-Ling Chang [22,29]. O’Neill et al. examined the effect of a 12-week multidisciplinary rehabilitation program in a cohort of esophagogastric cancer survivors in which they did not observe any significant improvements in HRQOL following the intervention. Of note, a significantly higher score in cognitive function was reported in the control group {F (1, 38) = 4.992, *p* = 0.031, η2 (eta squared) = 0.12} following the intervention [22]. In the second study by Yu-Ling Chang et al., delivering a 12-week nurse-led exercise and health education informatics program resulted in a significant improvement in HRQOL [29]. In this study, significant improvement in functioning scales, global health status, and symptom domains were reported in both the control and intervention groups (*p* < 0.01 to *p* < 0.001) and there were significant differences between groups regarding HRQOL domain scores at varied time points. For instance, in the intervention group, a significant improvement was observed in the functional domain at 1 month (β = 10.02, 95% CI 3.46, 16.58, *p* < 0.01) and 3 months (β = 10.25, 95% CI 1.55, 18.93, *p* < 0.05) after discharge while the role functioning score was significantly higher at all three time points (1 month: β = 11.68, 95% CI 0.90, 22.47, *p* < 0.05, 3 months: β = 14.03, 95% CI 2.43, 25.64, *p* < 0.05, 6 months: β = 16.12, 95% CI 4.21, 28.03, *p* < 0.01) following discharge. Significant improvements in emotional functioning were observed at 1-month post-discharge (β = 8.91, 95% CI 2.67, 15.15, *p* < 0.01) and in social functioning at 3 months after discharge (β = 9.25, 95% CI 0.20, 18.29, *p* < 0.05). Global QOL showed greater improvement in the intervention group 3 months following hospital discharge (β = 10.71, 95% CI 2.48, 19.94, *p* < 0.05). Regarding symptom domains, following GEE analysis, lower scores for insomnia at 1 and 3 months post-discharge (β = −14.50, 95% CI −22.91, −6.09, *p* < 0.01; and β = −12.81, 95% CI −2.74, −0.89, *p* < 0.05, respectively) were reported in the intervention group. Calculated scores for nausea and vomiting were also lower in the intervention compared to control groups at 3 and 6 months following discharge from hospital (β = −12.62, CI −20.48, −4.79, *p* < 0.01; and β = −11.67, 95% CI −20.77, −2.57, *p* < *0*.05, respectively) [29]. Analysis of esophageal cancer-specific quality of life scores (QOL-OES-18) showed lower scores for dry mouth (β = −8.68, 95% CI −16.86, −0.50, *p* < 0.05) at the one-month post-discharge assessment; similarly, there were lower scores for dysphagia at 3 months (β = −12.56, 95% CI −21.34, −3.76, *p* < 0.01) and at one and six months following discharge for loss of taste (β = −8.30, 95% CI −14.41, _2.19, *p* < 0.01; and β = −13.66, 95% CI −2240, −4.93, *p* < 0.01 respectively) in the intervention group versus the control [29].

### 3.2. Body Composition and Muscle Mass

Body composition, muscle mass, or anthropometric measurements were measured outcomes in three studies [22,28,29]. Body composition was measured by bioelectrical impedance analysis (BIA) in the studies by O’Neill and Yu-Juan Xu, and no significant differences in muscle mass were detected following the intervention [22,28]. There were inconclusive results regarding changes in anthropometric measurements. O’Neill et al. reported a higher mid-arm circumference in the intervention group at the post-intervention assessment (28.35 ± 4.70 compared to 28.55 ± 5.57, *p* = 0.019) and no changes in other anthropometric measurements of weight, BMI, and waist circumference. This was similar to the results of the Yu-Ling Chang study that reported no changes in body mass index. However, Yu-Juan Xu reported significantly less weight loss (2.7 kg, *p* < 0.001) following the intervention [22,28,29].

### 3.3. Physical Function and Cardiorespiratory Fitness

Functional capacity and cardiorespiratory fitness outcomes were examined in all five included studies [15,22,27,28,29]. Cardiorespiratory fitness was assessed in two studies by the cardiopulmonary exercise test (CPET) and both RCTs reported significant improvement in cardiorespiratory fitness following the rehabilitative intervention [22,29]. In the ReStOre study by O’Neill et al., the intervention group had a higher VO_2Peak_ compared to the control group (22.20 ± 4.35 versus 21.41 ± 4.49, *p* = 0.000) post-intervention as well as at the 3-month post-intervention assessment (VO_2 Peak_ 21.75 ± 4.27 versus 20.74 ± 4.65, *p* = 0.001) [22]. In the trial by Yu-Ling Chang et al., the intervention group had significantly higher VO_2max_ value post-intervention (β = 2.61, 95% CI 1.54, 3.69, *p* < 0.001) [29].

Three studies examined changes in functional capacity by the 6-min walk test (6 MWT) and a significant improvement was observed in all of them post-intervention [15,28,29]. Yu-Juan Xu et al. reported less decline in the 6 MWT walk distance following a walk and eat intervention during neoadjuvant chemoradiotherapy, compared to control (−18 ± 75 m vs. −118 m ± 160.5; group difference: 100 m, adjusted *p* = 0.012) [28]. Likewise, a significantly improved 6 MWT was observed (β = 83.30, 95% CI 52.60, 113.99, *p* < 0.001) following a rehabilitation intervention by Yu-Ling Chang et al. [29]. Moreover, another prehabilitation study showed an improvement in walking distance both before surgery (mean [SD] 6 MWD change, 36.9 [51.4] vs. −22.8 [52.5] m; *p* < 0.001) and after surgery in the intervention group versus control (mean [SD] 6 MWD change, 15.4 [65.6] vs. −81.8 [87.0] m; *p* < 0.001) [15].

Two studies measured hand-grip strength and both reported improvements following a multidisciplinary intervention. Yu-Juan Xu et al. observed a 3 kg less decrease in hand-grip strength in the intervention group (−1.1 ± 2.5 kg vs. −4.1 ± 4.0 kg, *p* = 0.002) and Ausania reported an increase of 4.8 and 5.9 kg in right and left-hand grip strength following prehabilitation, although this outcome was measured only in the intervention group [27,28].

### 3.4. Dietary Intake

None of the included studies reported dietary intake data. Although O’Neil et al. reported dietary assessment, dietary intake data were not reported in this study [22]. Likewise, Ausania and Minnella assessed nutritional intake and dietary habits before the intervention but no data regarding dietary intake was included in the results of these studies [15,27]. Similarly, Yu-Juan Xu reported an evaluation of dietary intake during weekly nutritional consult sessions, yet no dietary intake data was presented in this study [28].

### 3.5. Post-Op Complications/Other Outcomes

Three studies looked at complications post-surgery, and all reported no significant differences in the number or severity of complications, hospital stay, and readmission in the intervention group compared to standard care [15,27,28]. However, Yu-Juan Xu et al. observed lower rates of parenteral nutrition support (3.6% compared to 50%, *p* < 0.001) as well as a lower rate of wheelchair use (0% compared to 32.1%, (*p* < 0.01) in the intervention group [28]. Additionally, Yu-Ling Chang reported that the intervention group had significantly higher levels of albumin at 3 months post-discharge (β = 0.32, 95% CI 0.09, 0.54, *p* < 0.01) compared to the usual care group [29].

## 4. Discussion

This systematic review describes the effectiveness of multidisciplinary programs, whose core component was nutritional and exercise interventions, in improving outcomes in patients diagnosed with upper GI cancers. Nutritional interventions were mainly individual consults alone or along with prescribing oral nutritional supplements to ensure energy and protein requirements were met. It is noteworthy that the rationale behind prescribing nutritional supplement or support (PN) was not explained in the Ausania et al. and Minnella et al. studies. This is important as a lack of details regarding the delivered intervention may complicate interpretation of the reported results in these studies [15,27]. Exercise training included supervised and/or home-based walking and/or resistance training. Although there is growing interest in multidisciplinary care plans for cancer patients, acknowledging the potential beneficial effects of these interventions on patients’ quality of life, a limited number of studies have examined the effectiveness of these interventions in upper GI cancers to date [30,31,32].

Considering the results of the reviewed studies, interventions incorporating nutrition and exercise appear to be safe and acceptable in patients diagnosed with upper GI cancer, with a higher rate of compliance with supervised interventions [22,28]. However, only three studies reported compliance with the intervention, of which a single study reported adherence to nutrition consult sessions [15,22,28]. O’Neil et al. did not report adherence to dietetic consult and education sessions and Minnella et al. reported overall compliance with prehab intervention rather than specifying adherence to exercise and nutrition components separately [15,22]. A lack of sufficient data regarding the adherence rate in some of the reviewed studies makes it difficult to understand whether non-significant findings were due to ineffectiveness of the interventions or lack of compliance with the intervention. Additionally, reporting adherence to different components of a multidisciplinary intervention is important as it might help design acceptable and practical multimodal interventions. Moreover, it would be worthwhile to further examine factors that may play a role in patients’ enrolment and completion of these interventions, such as participants’ nutritional and psychological status, physical function, fatigue and weakness at baseline, transportation difficulties, and familial and professional commitments [33,34]. Acquiring more data on the underlying reasons for non-participation, withdrawal, and non-compliance is essential to design convenient, acceptable, and practical interventions.

Regarding physical functioning, the promising effects of combined nutritional interventions with physical training on functional and cardiorespiratory outcomes have been reported in several studies [35,36]. The included studies in this systematic review showed consistent results regarding improvements in patients’ physical functioning following multimodal interventions with core nutrition and exercise components. Whilst the reported functional outcomes varied among different studies, CPET and 6 MWT demonstrated an improvement following the intervention [15,22,28,29]. This improvement in physical performance is important as higher physical performance and functioning has been associated with improved quality of life, treatment response, and prognosis in cancer patients in several studies [37,38,39]. Although, further high-quality studies are needed to confirm the positive effects of multidisciplinary interventions on physical functioning due to the poor quality of the included studies.

The present systematic review showed that changes in body composition following multimodal interventions in upper GI cancer patients have been understudied and the limited available evidence showed mixed results. As mentioned previously, muscle wasting is associated with impaired quality of life and reduced survival following cancer treatments [40]. However, the optimal care that can effectively counteract cancer-associated cachexia has not been defined yet [40]. Cachexia and muscle wasting have received much attention in the scientific literature in recent years, but there remains limited evidence regarding the effectiveness of combined nutritional and exercise interventions in improving impaired body composition [16,41]. It is acknowledged that meeting patients’ protein and energy requirements is crucial to maintain muscle mass during cancer treatment and recovery, and exercise interventions may be effective in preventing and reversing muscle wasting [42,43,44]. It is worthy of note that, although four of the included studies assessed dietary intake at some time point of the study, none of them reported dietary intake data [15,22,27,28]. This could be considered as one of the factors that resulted in a high risk of bias in the reviewed studies as lack of dietary intake data makes it difficult to examine whether nutritional adequacy had been achieved by these interventions. Dietary intake assessment is crucial to ensure whether adequate protein and energy intake has been supported during multidisciplinary interventions. This is even more important in UGI cancer survivors as they experience severe reductions in their dietary intake and impaired food tolerance following surgeries [11].

The increased survival rate of patients diagnosed with cancer is accompanied by an acknowledgement of the need to ensure good quality of life. Cancer can negatively impact the quality of life and improved HRQOL may be associated with lower mortality and recurrence [45]. In the present review, changes in patients’ quality of life were assessed in two studies with inconsistent results. Assessing changes in HRQOL during and beyond cancer, and following interventions can assist researchers to have a better understanding of the effects of treatment on patients’ physical, mental, and emotional status [46].

The multidisciplinary interventions reviewed here failed to show any beneficial effects on post-operation complications and hospital stay. Although contradictory results were reported in other non RCT studies, with a lower mortality rate and severity of complications or shorter hospital stay following a multidisciplinary prehabilitation program [47,48], their small sample size and methodological limitations may limit their power to examine these outcomes properly. Optimization of patients’ nutritional and physical status before the surgery has been recommended for improving postoperative outcomes [49]. More high-quality research studies are needed to identify the most effective prehabilitation and/or rehabilitation interventions based on cancer site and type of surgery to improve operative outcomes.

Although this systematic review aimed to examine the optimal timepoint for delivery of multidisciplinary interventions within the cancer trajectory, a high risk of bias in the reviewed studies, and a high degree of variance regarding the design and delivery of the intervention and measured outcome prevented a clear conclusion on this issue. The limited studies that examined the effectiveness of prehabilitation versus rehabilitation in cancer reported inconsistent results from observing no significant differences in functional walking capacity to better responses to prehabilitaion compared with rehabilitation [50,51]. Therefore, the optimal timepoint for providing multidisciplinary care programs is yet to be known and requires further studies.

It should be acknowledged that the present systematic review has several limitations. First, although a comprehensive search was conducted to locate all relevant studies, some studies may have been missed. Secondly, a considerable number of ongoing trials were identified (12 relevant unpublished ongoing trials) and, when contacted, the authors of these studies reported that they were still ongoing, stalled due to the COVID-19 pandemic, or in pre-publishing stages [25]. The results of these studies may be helpful to yield a more explicit conclusion.

Moreover, it should be mentioned that the standard care varied significantly across different settings and in most settings, control groups were provided with standard care consisting of some nutrition and/or exercise prehabilitation or rehabilitation care, which may be considered as a confounding factor to determine the significant effects of these interventions on postoperative complications as well as nutritional status and quality of life. Furthermore, the importance of reporting details of standard care needs to be highlighted as it may complicate the interpretation of the results. For instance, it is unclear whether the lack of significant changes in body composition, HRQOL, and PA in the O’Neill study was due to the high quality of standard care as the details of standard care were not discussed [22].

On account of scarce evidence, and the variance and heterogenicity of studies in terms of intervention design, duration of study and follow-up, the time point of intervention delivery, and measured outcomes, etc., it was impossible to perform metanalysis and/or draw a firm conclusion on the effectiveness of these interventions on measured outcomes.

Adding to this, as indicated in Table 3, the primary outcome varied significantly across the included studies, which may limit meaningful comparisons across studies and evidence synthesis. Moreover, as primary outcomes are the basis of conducting power and sample size calculations, some studies may not be adequately powered to detect the group differences for a specific outcome of interest in this systematic review. For instance, the Yu-Juan Xu et al. study has a low power (48%) for detecting group differences in lean muscle mass [28].

Finally, as the results of the risk of bias assessment indicated, more high-quality and well-designed RCTs are required to draw more robust judgment on the effectiveness of these multidisciplinary interventions, specifically on body composition and quality of life. Some key points need to be considered to reduce the risk of bias in future studies, including considering blinded assessment and analysis, dietary intake assessment, and reporting dietary intake data to ensure nutritional adequacy, providing greater details of delivered intervention, such as the rationale behind prescribing nutrition support or supplements, providing greater details regarding adherence to different components of the intervention, and describing details of standard/control care.

## 5. Conclusions

Limited evidence is available on the effectiveness of multimodal interventions, with a core component of nutrition and exercise, in improving outcomes in upper GI cancer patients. Although studies showed an improvement in physical function and exercise capacity, the evidence regarding positive changes in muscle mass and quality of life was scarce and conflicting. In conclusion, due to the poor quality of limited available evidence, further high-quality studies are warranted to examine the effectiveness of multidisciplinary care programs in improving outcomes in upper GI cancers with a focus on improving the body composition and quality of life in these patients. Considering a core outcome set for measuring the clinical effectiveness of multidisciplinary supportive care programs may improve the consistency and quality of future investigations [52]. Moreover, it may facilitate more meaningful comparison across studies and support evidence synthesis.

## Figures and Tables

**Figure 1 nutrients-13-02842-f001:**
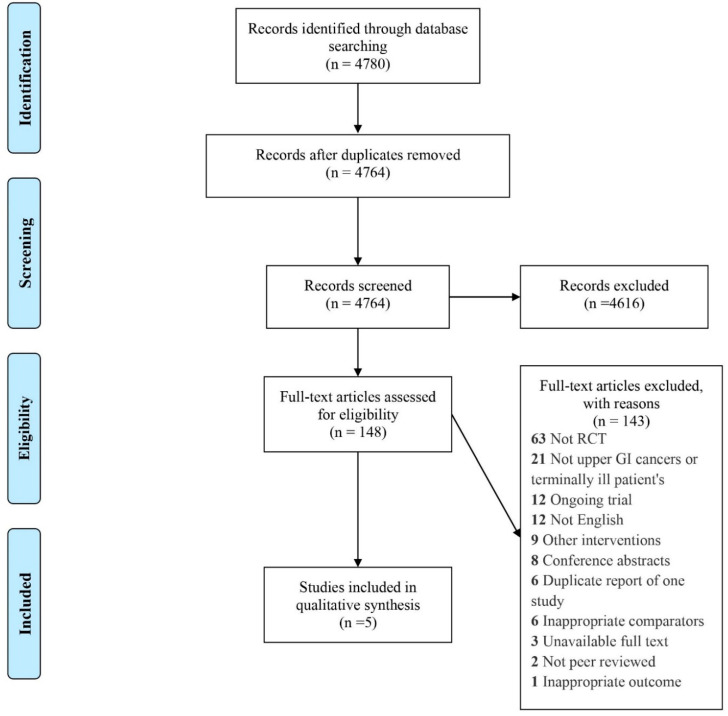
PRISMA flow diagram.

**Figure 2 nutrients-13-02842-f002:**
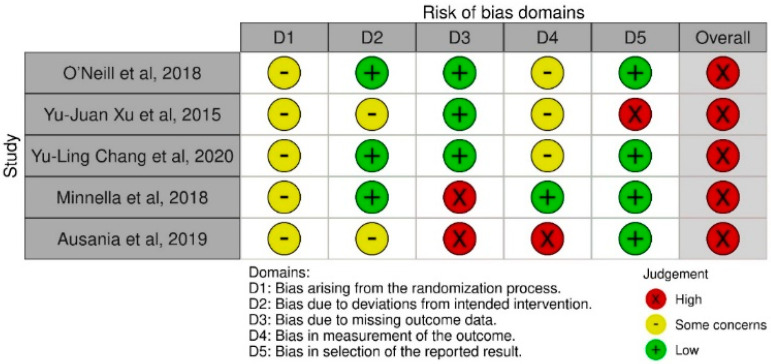
Risk of bias assessment. Risk of bias completed using the revised Cochrane Risk of Bias tool (RoB2).

**Table 1 nutrients-13-02842-t001:** Overview of the included studies.

Author, Year	O’Neill et al., 2018	Yu-Juan Xu et al., 2015	Yu-Ling Chang et al., 2020	Minnella et al.,2018	Ausania et al., 2019
Country	Ireland	Taiwan	Taiwan	Canada	Spain
Cancer type	Esophagogastric	Esophageal	Esophageal	Esophagogastric	Pancreaticoduodenal
Timepoint of Intervention	Rehab (6 mo–5 yrs. Post-treatment)	During neoadjuvant chemoradiotherapy	Rehab (Immediately after surgery)	Prehab	Prehab
Participant No (Randomized)	43 (Intervention = 21, Control = 22)	59 (Intervention = 30, Control = 29)	88 (Intervention = 44, Control = 44)	68 (intervention = 34, control = 34)	40 (intervention = 18, Control = 22)
Participants’ Sex (%Male)	81%	92.9%	90.9%	74.5%	55%
Participants’ Age	Intervention: 67.19 ±7.49 Control: 64.14 ± 10.46	Intervention:58.1 ± 9.6 Control:61.1 ± 9.0	Intervention:56 ± 8.9 Control: 56 ± 10.0	Intervention:67.3 ± 7.4 Control:68.0 ± 11.6	Intervention:66.1 (38–80) Control: 65.7 (38–81)
Recruitment rate	40.3%	92%	94.6%	60.1%	96%
Dropout rate	9.3%	5.1%	9.1%	25%	16%
Adherence	Supervised exercise: 94 ± 12% Unsupervised exercise:78 ± 27	Nutritional sessions:100% Walking sessions: 68% (32%–100%). Target maximal heart rate was achieved in 71% 54% completed more than 80% of walking sessions	NR	63%	NR
Adverse event	None	NR	NR	None	NR

Abbreviations: NR Not reported.

**Table 2 nutrients-13-02842-t002:** Description of interventions in included studies.

Author, Year	O’Neill et al., 2018	Yu-Juan Xu et al., 2015	Yu-Ling Chang et al., 2020
Exercise component	14 Supervised aerobic training (treadmill walking, stationary cycling, cross-training) 14 Supervised resistance training (free weights, horizontal leg press) 37 unsupervised home-based aerobic (walking or stationary cycling) 10 unsupervised resistance training (using TheraBand)	Nurse-supervised walking three times per week 5-min warm-up (ankle circles, leg swings, pelvic loops, arm circles), 20 min of hallway ambulation at the patient’s own pace before or after radiotherapy Intensity of 60%	Home-based walking at a moderate intensity level after meals, 3–5 days per week for 30 min, or a total of 150 min per week Desired heart rate reserve percentage: 55–65%.
Nutrition component	Nutritional assessment 1:1 Dietary counselling Personalized nutritional advice Number of sessions depended on patients’ status	Weekly dietetic consult (weight and intake assessment, advice on eating and feeding difficulties, food or formula selection, skills for modifying food texture, and oral care before and after eating	E-Books containing dietary guidance and advice
Other components	7 group education sessions delivered by surgeon, dietitian, physiotherapist, occupational therapist, psychotherapist specialized in mindfulness	None	e-books (exercises, symptom management and psychological advice) Online nurse support to answer survivors’ questions An online discussion group referring patients to appropriate medical professionals for an in-person visit if needed
Control	Standard care (standard clinical care as per best practice)	Standard care (nutritional and self-care advice from nurses)	Standard care (conventional postoperative feeding, wound care, and regular postoperative rehabilitation exercises)
Duration	12 W	4–5 W	12 W
**Author, Year**	**Minnella et al., 2018**		**Ausania et al., 2019**
Exercise component	Home-based aerobic exercise (brisk walk, jogging, or cycling) 3 days per week,30 min each day (including 5-min warm-up and 5-min cooldown) Strengthening activity, 3 sets of 8 to 12 repetitions for 8 muscle groups using TheraBand, 1 day/week, 30 min (including 5-min flexibility and 5-min stretching) a weekly telephone call by a kinesiologist		5 sessions (60 min each) high-intensity endurance training performed on a cycle-ergometer stationary bicycle, 10 min warm-up cycling, 20 min muscle toning exercise, 20 min aerobic exercise, 10 min cool-down. Unsupervised home-based functional exercises and breathing exercises
Nutrition component	Nutritional assessment and consult Whey protein supplement if required Weekly phone call by a dietitian		Nutritional assessment Nutritional support (liquid oral nutrition supplements and vitamin supplements.) Total parenteral nutrition if required Follow-up in the outpatient clinic
Other components	None		Metformin or insulin if required for BS control Blood glucose monitor to check glucose level at home Pancreatic enzymes replacement therapy
Control	Standard care (perioperative care according to the ERAS Society Guideline protocol, + Nutritional counselling session)		Standard care (nutritional counselling, physical activity recommendation and advice on smoking cessation and if indicated pancreatic enzyme supplementation, dietitian referral, preoperative biliary drainage)
Duration	Median 36 days (IQR, 17–73)		Median 12.6 days (Minimum 7 days was planned in intervention)

Abbreviations: W weeks, ERAS enhanced recovery after surgery, IQR interquartile range, BS blood sugar.

**Table 3 nutrients-13-02842-t003:** Summary of results.

Author, Year	Relevant Outcome Measure	Results
O’Neill et al., 2018	Cardiorespiratory fitness (CPET, VO2 Peak) * HRQOL (EORTC-QLQ-C30) Body composition &anthropometric measurements (BIA, weight, height, waist and midarm circumference) Physical activity level (Actigraph GT3Xþ) Dietary intake (24 h food recall)	Significant improvement in cardiorespiratory fitness post-intervention (VO2 Peak:22.20 ± 4.35 versus 21.41 ± 4.49, *p* = 0.000) and 3 months post-intervention (VO2 Peak 21.75 ± 4.27 versus 20.74 ± 4.65, *p* = 0.001) No significant changes in HRQOL, except cognitive function higher in the control group following the intervention [100.00 ± 16.67 compare to 83.33 ± 16.67, *p* = 0.031] No changes in anthropometric measurements and body composition except higher mid-arm circumference in the intervention group at post-intervention assessment (28.35 ± 4.70 compare to 28.55 ± 5.57, *p* = 0.019) No changes in PAL Not reported
Yu-Juan Xu et al., 2015	Functional capacity (6 MWT, HGS) * Body composition &anthropometric measurements (BIA, weight) * Treatment tolerance (chemotherapy or radiotherapy interruption, unplanned hospital admission, grade > 2 neutropenia, fever > 38.5 °C, intravenous nutritional support, wheelchair use)	100 m less decline in walk distance (*p* = 0.012) 3 kg less decrease in hand-grip strength (*p* = 0.002) Non-significant less muscle mass loss (1.3 kg, *p* = 0.057) Significant less weight loss (2.7 kg, *p* < 0.001) No differences in chemotherapy/radiotherapy interruption, unplanned hospital admissions, neutropenia, fever Lower rates of parenteral nutrition support (3.6% compared to 50%, *p* < 0.001) Lower rate of wheelchair uses (0% compared to 32.1%, (*p* < 0.01)
Yu-Ling Chang et al., 2020	Exercise capacity (CPET (VO2 max), 6 MWT) HRQOL (EORTC-QLQ-C30 & EORTC QLQ-OES18) * Albumin Daily steps (smart bracelet) BMI	Higher maximal oxygen consumption (β = 2.61, 95% CI 1.54, 3.69, *p* < 0.001), effect size = 0.97 Greater distance on the six-minute walking test (β = 83.30, 95% CI 52.60, 113.99, *p* < 0.001), effect size = 0.36 Significantly better HRQOL (EORTC-QLQ-C30) at different time points following discharge. Physical (1 and 3 months), role (1, 3, and 6 months), emotional (1 month), social (3 months) and global health (3 months), insomnia (1 and 3 months) and nausea/vomiting (3 and 6 months). Improvement in oesophageal cancer-specific symptoms (EORTC QLQ-OES18), dry mouth (1 month), dysphagia (3 months), loss of taste (1 and 6 months) Higher levels of albumin at 3 months after discharge (β = 0.32, 95% CI 0.09, 0.54, *p* < 0.01), minimal effect size Data not reported No changes in BMI
Minnella et al., 2018	Functional capacity (6 MWT) * Post-operative morbidity (CDC &CCI), length of hospital stay, 30-day hospital visit, readmission rate, death, adherence to planned neoadjuvant therapy	Improvement in functional capacity before (mean [SD] 6 MWD change, 36.9 [51.4] vs. −22.8 [52.5] m; *p* < 0.001) and after surgery (mean [SD] 6 MWD change, 15.4 [65.6] vs. −81.8 [87.0] m; *p* < 0.001). No differences were observed in post-portative outcomes
Ausania et al., 2019	Cardiopulmonary status (FEV 1, FVC, O2 sat%) Functional capacity (10 MWT &HGS) Post-operative complications (CDC) *, pancreatic leak (type B&C), DGE, HS, readmission	Improvement respiratory function in the intervention group compared to baseline, changes in FVC (l, median) = 0.6, FEV 1 (l, median) = 0.48, O2 sat% = 0.3 Improvement in functional capacity in the intervention group compared to baseline 10-m walk test (1.2 s), HGS (left hand = 5.9, right hand = 4.8) No differences in post-op complications, hospital stay, readmission between intervention and control group, Significantly lower DGE, 5.6% vs. 40.9% in the standard care group (*p* = 0.01)

Abbreviations: CPET Cardiopulmonary Exercise Testing, VO2 Peak highest value of oxygen uptake, HRQOL Health-Related Quality Of Life, EORTC-QLQ-C30 The European Organization for Research and Treatment of Cancer Quality of Life questionnaire-Cancer, BIA Bioelectrical Impedance Analysis, 6 MWT Six-Minute Walk Test, HGS Hand Grip Strength, VO2 max Maximum rate of oxygen uptake, EORTC QLQ-OES18 EORTC esophageal cancer-specific questionnaire, BMI Body Mass Index, CDC Clavien-Dindo classification, CCI Comprehensive Complication Index, FEV 1 forced expiratory volume in one second, FVC forced expiratory volume, O2 sat oxygen saturation, 10 MWT 10 Meter Walk Test, DGE delayed gastric emptying, HS Hospital Stay. * Primary outcome/s of the study.

## Data Availability

No new data were created or analyzed in this study. Data sharing is not applicable to this article.

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
