# Peer review of "The Effectiveness of Nutrition Interventions Combined with Exercise in Upper Gastrointestinal Cancers: A Systematic Review"

_nutrients, 2021, doi:10.3390/nu13082842_

Round 1
Reviewer 1 Report
This is a well written systematic review on nutrition and exercise interventions in people with upper gastrointestinal cancer. An important area of significant interest to health professionals and researchers. However, I have some comments for the authors to address:
- The outcomes reported in the paper don't fully align with the outcomes that were planned according to the registered protocol in PROSPERO, in particularly changes in dietary intake which is not addressed at all. If this wasn't assessed in the included studies this should be reported.
- In Table 1, an adherence of 63% is reported for the study by Minella et al. It is not clear which component of the intervention this refers to - exercise, nutrition, both? This needs some further detail or if this level of detail is not reported in the original paper this should be specified.
- Table 2, no details of standard care are described for the study by O' Neill. The description of standard care is important to understand when interpreting the results, for example if the quality of standard care is already high then there may be limited benefit that can be achieved with a multimodal program. If standard care was not described this should be reported.
- Table 2, there are 2 studies where it is stated as part of the nutrition intervention that PN (Ausania et al) or whey protein supplements (Minella et al) were used if required. Were any details provided in these studies of how this decision was made, e.g. if nutritional intake reduced to less than 50% of requirements? This is also important context regarding how the intervention was delivered that can affect the interpretation of the results.
- The paper would benefit from an additional table that describes the relevant outcomes that were assessed in each included study and a summary of the results. Although the results are reported in text, a summary in a table would be really helpful.
- In the results, body composition findings are reported under health-related quality of life yet body composition is listed as the first outcome of interest in the methods and the protocol. This outcome should really have it's own heading in the results.
- While only a small number of studies were located which doesn't lend itself to any detailed sub-analysis it is clear that some interventions were delivered as prehab and some as rehab. Could the authors briefly comment on whether there appeared to be any difference in the findings between prehab versus rehab focused intervention. This is relevant given prevention of deterioration is often easier to achieve than treating deterioration after the fact.
- In the discussion, page 11 lines 292 - 303 where body composition findings are discussed there is mention of the importance of meeting protein and energy requirements however, this is not addressed in the results at all. As per my earlier comment assessment of dietary intake did seem to be an intended outcome but has been omitted. if this was due to studies not reporting this outcome that is an important findings to report in itself and is a flaw in the studies. it is difficult to know if nutritional adequacy has been achieved without assessing dietary intake.
- Similarly to the above comment, in the studies where adherence to the intervention was not reported if these studies reported negative findings we wouldn't know if this is due to ineffectiveness of the intervention or if the intervention was not actually received as intended. Can the authors acknowledge this and comment?
Author Response
Dear reviewer,
Many thanks for your kind comments and questions. In response to your comments the following explanations and replies (in red) are provided:
- The outcomes reported in the paper don't fully align with the outcomes that were planned according to the registered protocol in PROSPERO, in particularly changes in dietary intake which is not addressed at all. If this wasn't assessed in the included studies this should be reported.
Dietary intake data was not unfortunately reported in the included studies. We have included dietary intake in the outcomes and results section of the revised manuscript and have also expanded the discussion to highlight this absence of reported dietary outcomes in multidisciplinary interventions.
- In Table 1, an adherence of 63% is reported for the study by Minella et al. It is not clear which component of the intervention this refers to - exercise, nutrition, both? This needs some further detail or if this level of detail is not reported in the original paper this should be specified.
In the original paper, the authors reported overall compliance with prehabilitation intervention and did not specify the adherence to exercise or nutrition component separately. This issue has been discussed in the revised manuscript Discussion. (Page 12, Line 299-305)
- Table 2, no details of standard care are described for the study by O' Neill. The description of standard care is important to understand when interpreting the results, for example if the quality of standard care is already high then there may be limited benefit that can be achieved with a multimodal program. If standard care was not described this should be reported.
Details of standard care were not explicitly described by O’Neill. Usual care was described as standard clinical care as per best practice and further details were not discussed. This has been reported and acknowledged in the revised manuscript. (Page 13, Line 376-380)
- Table 2, there are 2 studies where it is stated as part of the nutrition intervention that PN (Ausania et al) or whey protein supplements (Minella et al) were used if required. Were any details provided in these studies of how this decision was made, e.g. if nutritional intake reduced to less than 50% of requirements? This is also important context regarding how the intervention was delivered that can affect the interpretation of the results.
Minella et al reported that Whey protein supplements were prescribed to guarantee a daily protein intake of 1.2 to 1.5g/kg of ideal body weight (or approximately 20% of total energy requirements). Unfortunately, there were not any details regarding the guidelines for that PN that were prescribed by Ausania et al. This issue and its probable effects on the quality of the study has been discussed in the revised manuscript. (Page 12, Line 287-289)
- The paper would benefit from an additional table that describes the relevant outcomes that were assessed in each included study and a summary of the results. Although the results are reported in text, a summary in a table would be really helpful.
The authors agree that a summary table is beneficial, and this has been added to the revised manuscript as Table 3. (Page 9-10)
- In the results, body composition findings are reported under health-related quality of life, yet body composition is listed as the first outcome of interest in the methods and the protocol. This outcome should really have it's own heading in the results.
The body composition had its own heading in the result section (3.2. Body Composition and Muscle Mass, Line 255 in original manuscript and line 229 in the revised manuscript)
- While only a small number of studies were located which doesn't lend itself to any detailed sub-analysis it is clear that some interventions were delivered as prehab and some as rehab. Could the authors briefly comment on whether there appeared to be any difference in the findings between prehab versus rehab focused intervention. This is relevant given prevention of deterioration is often easier to achieve than treating deterioration after the fact.
Due to the poor quality of reviewed study and a high degree of variance regarding design and delivery of the intervention, measured outcome and some significant differences in baseline characteristics of intervention and control group, commenting on the effectiveness of prehab versus rehab was not possible. This issue has been discussed in the revised manuscript. (Page 13, Line 357-364)
- In the discussion, page 11 lines 292 - 303 where body composition findings are discussed there is mention of the importance of meeting protein and energy requirements however, this is not addressed in the results at all. As per my earlier comment assessment of dietary intake did seem to be an intended outcome but has been omitted. if this was due to studies not reporting this outcome that is an important finding to report in itself and is a flaw in the studies. it is difficult to know if nutritional adequacy has been achieved without assessing dietary intake.
As mentioned in response to comment 1, none of the included studies reports dietary intake. This has been reported and discussed in the revised manuscript. (Page 12, Line 332-340)
- Similarly, to the above comment, in the studies where adherence to the intervention was not reported if these studies reported negative findings we wouldn't know if this is due to ineffectiveness of the intervention or if the intervention was not actually received as intended. Can the authors acknowledge this and comment?
The authors acknowledge that this is a very important point. Some of the studies did not report the adherence rate or did not provide sufficient details on adherence to different components of the intervention. This further adds to the fact that all of the included studies had high risk of bias and there is a need to conduct high-quality clinical trials in the future with robust reporting. These issues have been discussed in the revised manuscript. (Page 12, Line 297-305)
Reviewer 2 Report
I think this paper is well written and very topical. Some minor changes are recommended:
- introduction mentions cachexia and sarcopenia, but only discusses muscle mass/changes in muscle mass with no mention/explanation around fat mass and sarcopenia - hence it is recommended to either take sarcopenia out, or ensure both sarcopenia and cachexia are both discussed.
- Search strategy (second paragraph 2.2.7) says there was no limit on language but Figure 1 states excludes non-english articles.
- Could Table 1 and 2 be combined.
- Formatting for 3.2 title
- Results 3.1-3.3 are extensive challenging to read - could there be another way to represent this data. Also a summary at the end of each section incorporating the quality of the evidence would be useful.
- Discussion should have greater emphasis on the poor quality of the research when discussing results.
- The last paragraph in the discussion that does acknowledge risk of bias would be strengthened by including some key aspects that need to be considered when planning an intervention in order to reduce risk of bias
Author Response
Dear reviewer,
Many thanks for your kind comments and questions. In response to your comments following explanations and replies are provided:
1.introduction mentions cachexia and sarcopenia, but only discusses muscle mass/changes in muscle mass with no mention/explanation around fat mass and sarcopenia - hence it is recommended to either take sarcopenia out, or ensure both sarcopenia and cachexia are both discussed.
Cachexia and muscle loss has been discussed in the revised version and sarcopenia has been taken out.
2.Search strategy (second paragraph 2.2.7) says there was no limit on language, but Figure 1 states excludes non-english articles.
To clarify, there was no limit on language during databases search, although non-English manuscripts have been excluded in the full-text review stage.
3.Could Table 1 and 2 be combined.
The authors acknowledge the reviewer’s suggestion. We have tried different designs to combine Table 1&Table 2, but unfortunately, it was not easy to read due to the extensive amount of information. In the interest of the reader, we have kept them separated.
4.Formatting for 3.2 title
Formatting has been edited in the revised version.
5.Results 3.1-3.3 are extensive challenging to read - could there be another way to represent this data. Also, a summary at the end of each section incorporating the quality of the evidence would be useful.
A table including measured outcomes in each study and a summary of results has been added to the result section. (Table 3, Page 9-10). The quality of the evidence regarding each outcome has been discussed in the relevant section of the discussion in the revised version.
6. Discussion should have a greater emphasis on the poor quality of the research when discussing results.
Revised mention of the manuscript further discussed the poor quality of available evidence. For instance, the following factors that may result in high risk of bias in the reviewed studies have been discussed:
Lack of sufficient details regarding adherence to the intervention and its different components (Page 12, Line 297-305)
Lack of reporting dietary intake data (Page 12, Line 332-340)
Lack of reporting details of standard care (Page 13, Line 376-380).
7.The last paragraph in the discussion that does acknowledge risk of bias would be strengthened by including some key aspects that need to be considered when planning an intervention in order to reduce risk of bias.
A brief discussion of key points which need to be considered in future studies to improve the quality of the evidence has been added to the discussion. (Page 13, Line 387-392).
Round 2
Reviewer 1 Report
Thank you to the authors for their response to my comments and revisions to the manuscript. All my comments have been addressed and I have no further suggestions to make.
Author Response
Dear reviewer,
The authors would like to thank you for your helpful comments and suggestions, which improved the manuscript to a great extent.